

# Association of weaning preparedness with extubation outcome of mechanically ventilated patients in medical intensive care units: a retrospective analysis

Feng-Ching Lin[1,2], Yao-Wen Kuo[1], Jih-Shuin Jerng[3,4] and Huey-Dong Wu[1]

[1] Department of Integrated Diagnostics & Therapeutics, National Taiwan University Hospital, Taipei, Taiwan
[2] Department of Nursing, Cardinal Tien Junior College of Healthcare and Management, New Taipei City, Taiwan
[3] Department of Internal Medicine, National Taiwan University Hospital, Taipei, Taiwan
[4] Center for Quality Management, National Taiwan University Hospital, Taipei, Taiwan

Corresponding author
Jih-Shuin Jerng, jsjerng@ntu.edu.tw

## ABSTRACT

**Background:** Assessment of preparedness of weaning has been recommended before extubation for mechanically ventilated patients. We aimed to understand the association of a structured assessment of weaning preparedness with successful liberation.

**Methods:** We retrospectively investigated patients with acute respiratory failure who experienced an extubation trial at the medical intensive care units of a medical center and compared the demographic and clinical characteristics between those patients with successful and failed extubation. A composite score to assess the preparedness of weaning, the WEANSNOW score, was generated consisting of eight components, including *W*eaning parameters, *E*ndotracheal tube, *A*rterial blood gas analysis, *N*utrition, *S*ecretions, *N*euromuscular-affecting agents, *O*bstructive airway problems and *W*akefulness. The prognostic ability of the WEANSNOW score for extubation was then analyzed.

**Results:** Of the 205 patients included, 138 (67.3%) patients had successful extubation. Compared with the failure group, the success group had a significantly shorter duration of MV before the weaning attempt ($11.2 \pm 11.6$ vs. $31.7 \pm 26.2$ days, $p < 0.001$), more with congestive heart failure (42.0% vs. 25.4%, $p = 0.020$), and had different distribution of the types of acute respiratory failure ($p = 0.037$). The failure group also had a higher WEANSNOW score ($1.22 \pm 0.85$ vs. $0.51 \pm 0.71$, $p < 0.001$) and worse Rapid Shallow Breathing Index ($93.9 \pm 63.8$ vs. $56.3 \pm 35.1$, $p < 0.001$). Multivariate logistic regression analysis showed that a WEANSNOW Score = 1 or higher (OR = 2.880 (95% CI [1.291–6.426]), $p = 0.010$) and intubation duration >21 days (OR = 7.752 (95% CI [3.560–16.879]), $p < 0.001$) were independently associated with an increased probability of extubation failure.

**Conclusion:** Assessing the pre-extubation status of intubated patients in a checklist-based approach using the WEANSNOW score might provide valuable

insights into extubation failure in patients in a medical ICU for acute respiratory failure. Further prospective studies are warranted to elucidate the practice of assessing weaning preparedness.

# INTRODUCTION

Weaning from mechanical ventilation (MV) is an essential task in the care of patients with acute respiratory failure. Despite advances in assessing the feasibility of extubation, the rate of reintubation remains high at 10–19%, with a high mortality rate in those who fail extubation of 25–50% (*Thille, Cortés-Puch & Esteban, 2013*). The failure rate at the first attempt to wean can be as high as 50% (*Peñuelas et al., 2011*) and even up to 70% for those ventilated for more than 2 days (*Tonnelier et al., 2011*). Previous studies have reported the application of various assessments and predictors to increase successful weaning and extubation. Factors associated with extubation failure include age, the primary reason for intubation, neurological dysfunction, cough efficacy and amount of secretion (*Tonnelier et al., 2011*). Extubation failure may also be caused by hidden factors such as delirium (*Ely et al., 2004*) and intensive care unit (ICU)-acquired weakness (*Garnacho-Montero et al., 2005*).

Spontaneous breathing trials are currently the fastest approach to wean patients (*MacIntyre et al., 2001*), and it has been reported to be the most accurate predictor of extubation failure (*Yang & Tobin, 1991*). A single daily spontaneous breathing trial lasting from 30 min to 2 h has been recommended as the primary approach to weaning (*Cook et al., 2000*; *MacIntyre et al., 2001*). However, the probability of successful extubation in patients with acute respiratory failure may be related to a variety of factors not limited to traditional "weaning parameters", and a clinical composite scoring method is lacking. A possible approach is to assess the preparedness of weaning by using a checklist-based bundle approach, such as the WEANSNOW checklist, which has been recommended to assess the degree of preparedness of a patient to begin breathing spontaneously (*Hasani & Grbolar, 2008*; *Witt, 2008*; *Frances, 2010*; *Godara et al., 2014*; *Johnson & Abraham, 2015*; *Mendez, 2018*). This evaluation method involves the assessment of eight items. However, the WEANSNOW checklist has not previously been investigated with regards to predicting the success of liberation from MV. Therefore, this study aimed to assess the prognostic ability of a composite score based on the WEANSNOW checklist for extubation failure in intubated and mechanically ventilated patients in a medical intensive care unit.

# MATERIALS AND METHODS

## Study design

This retrospective analysis was conducted at the medical ICUs of National Taiwan University Hospital, a university-affiliated medical center in northern Taiwan.

The Institutional Research Ethics Committee A of the National Taiwan University Hospital approved this study (#201808101RINA) and waived the need for informed consent from the patients.

## Setting

The medical ICUs in this hospital contain a total of 57 beds, with more than 90% bed occupancy; more than 70% of the admitted patients are given MV support due to acute respiratory failure. A physician-initiated, protocolized weaning process is used in these ICUs, in which the attending physician, who is a certified intensivist, decides whether the patient's condition is suitable for a progressive reduction in the level of MV support by daily screening for the preparedness of a spontaneous breathing trial. The first weaning trial for an intubated patient is performed when they meet the following criteria: resolution of the disease for which the patient was intubated, cardiovascular stability without or minimal vasopressors, no continuous sedation and adequate oxygenation defined as a $PaO_2/FiO_2$ ratio of at least 150 mm Hg with positive end-expiratory pressure of up to 8 cm $H_2O$ (*Boles et al., 2007*). The practice of respiratory therapists was to perform physiological assessment and provided the results as "weaning parameters" to the medical records for most of the patients, whereas the attending intensivist decided to extubate the patients. The ICUs of this study did not mandate the documentation of WEANSNOW assessment on the medical records. Extubation is then performed based on the conventional approach of a successful single daily spontaneous breathing trial lasting from 30 min to 2 h. If the trial is unsuccessful, the patient is allowed to rest before another trial is attempted (*MacIntyre et al., 2001*). This study focused on patients who underwent extubation based on this standardized weaning protocol. The patients then underwent extubation and liberation from MV and their general and respiratory status were observed for at least 48 h. The clinical care team then decided whether the successfully extubated patients could be transferred out of the ICU.

## Participants

We screened the electronic medical records of patients with acute respiratory failure admitted to our medical ICUs from July 2017 to December 2018 for the eligibility of inclusion into this study. Patients who had undergone at least one spontaneous breathing trial followed by extubation were identified and enrolled, including those with documented extubation failure, defined as the need for reintubation with mechanical or noninvasive ventilation or death due to respiratory failure without re-intubation, within 2 days after extubation (the "failure" group) (*Rothaar & Epstein, 2003*), and those who were successfully extubated during the same period (the "success" group). Of the patients with multiple extubation attempts during the same ICU stay, only the first attempt was included for analysis. Patients were excluded if they had at least one of the following conditions: documented endotracheal tube withdrawal and MV as an end-of-life measure, inadvertent removal of the endotracheal tube despite a previous weaning trial and patients with missing data related to the variables required for analysis.
**Table 1 Components of the proposed WEANSNOW bundle evaluation.**

| Acronym | Component | Description[*] |
|---|---|---|
| $W$ | Weaning profile | Pre-extubation physiological profile meets all of the following criteria: MIP = −30 cm $H_2O$ or better; MEP = +30 cm $H_2O$ or better; spontaneous $V_T \geq 5$ ml/IBW/kg; spontaneous $V_E \leq 10$ L/m; RSBI < 105 breaths/min/L |
| $E$ | Endotracheal tube | Endotracheal tube with internal diameter ≥ 7.0 mm |
| $A$ | Arterial blood gas data | $P_aO_2 \geq 60$ mmHg; no metabolic alkalosis |
| $N$ | Nutrition | Albumin ≥ 3.0 gm/dl; no electrolyte derangement |
| $S$ | Secretion | Airway suctioning ≤ 1 session per hour |
| $N_2$ | Neuromuscular condition | Not under neuromuscular depressing medications |
| $O$ | Obstruction of main airways | Clinical absence of airway obstruction, assessed by breathing sounds, chest radiography, ventilator display, and cuff deflation leak percentage ≥ 15.5% |
| $W_2$ | Wakefulness | Glasgow Coma Scale ≥ 12 |

**Note:**

[*] MIP, maximum inspiratory pressure; MEP, maximum expiratory pressure; $V_T$, tidal volume; $V_E$, minute ventilation; RSBI, rapid shallow breathing index.

## Development of the WEANSNOW bundle assessment score

The operational definitions of the individual components of the WEANSNOW score (WS) was decided after a discussion among the investigators based on the literature regarding suitability for weaning (*Hasani & Grbolar, 2008*; *Witt, 2008*; *Frances, 2010*; *Godara et al., 2014*; *Johnson & Abraham, 2015*; *Mendez, 2018*). Eight components were identified: $W$ (weaning profile), $E$ (endotracheal tube), $A$ (arterial blood gas analysis), $N$ (nutrition), $S$ (secretions), $N_2$ (neuromuscular), $O$ (obstruction of the airway), $W_2$ (wakefulness). By reviewing the literature, we found a lack of evidence-based generation of the WEANSNOW criteria, despite that this score had appeared in several references. The author decided each criterion for the individual components based on the clinician experience-based consensus among the four investigators of this study. These components were regarded to be modifiable components to assess the feasibility of testing spontaneous breathing and final extubation. For the Glasgow Coma Score of the "Wakefulness" item, the verbal component was designated as five if the patients could provide a correct response to the assessing nurse through any non-verbal communication. This item was scored as 1 if the patient did not provide any non-verbal response to the assessing nurse, whereas the patient would be scored as 3 if the patient showed a response but not considered correct to the nurse's calling or direction. As this study considered that the concept of "preparedness" should be given to the criteria, each bundle component was scored 0 if the patient fulfilled the operational definition of the assessment or 1 if they did not. The scores of all eight components were summed for the WS composite score, with a minimum of 0 and a maximum of 8 points. Definitions of the operational criteria are summarized in Table 1. If there were more than one session of clinical and physiological assessment regarding the WEANSNOW components, only the assessment data closest to the time of extubation would be used.
## Data collection and data sources

The investigators collected clinical information from the institutional electronic medical record system, including gender, age, admission diagnosis, comorbidities, units, admission APACHE II score, length of MV use and ICU length of stay. As this study was retrospective, the investigators retrieved the individual information regarding the components of the WS from the electronic medical records.

## Statistical analysis

All statistical analyses were carried out using STATA statistical software (StataCorp LLC, College Station, TX, USA). We first performed a descriptive analysis of the background of the patients. Clinical and demographic characteristics, physiological parameters related to weaning and extubation assessments, components of the WS and total WSs were compared between the failure and success groups. We then performed multivariate logistic regression analysis, including significant variables in univariate analysis, defined as a $p$-value of 0.1 or less in the univariate analysis comparing the success and failure groups, to assess the potential factors associated with weaning outcomes.

Continuous variables were expressed as mean ± SD and categorical variables as number and percentage. Inferential statistics were performed using the chi-square test, Fischer's exact test, and independent $t$-test. A $p$-value of < 0.05 was considered to be statistically significant.

# RESULTS

## Participants

Figure 1 shows the flow diagram of the patients included in this study. A total of 289 patients admitted were supported with mechanical ventilation during the study period and 205 of them underwent at least one attempt of extubation at the ICU, including 138 (67.3%) who succeeded in the first attempt of extubation and 67 (32.7%) who failed. For the failure group, 22 were re-intubated without receiving noninvasive ventilation, whereas 38 received NIV, but 28 were later reintubated and 10 were liberated from NIV upon transferred out the ICU; seven patients died because of withholding reintubation or noninvasive ventilation.

Regarding the performance of the ICU in this study, there were 321 admitted patients during the study period, including 289 receiving MV. Of these 289 patients, 148 (51.2%) were transferred out of the ICU after successful liberation from MV, 91 (31.5%) died and 50 (17.3%) were transferred out of the ICU under a ventilator-dependent state.

Table 2 compares the demographic and clinical characteristics between the two groups. Compared with the failure group, the success group had a significantly shorter duration of MV before the weaning attempt (11.2 ± 11.6 vs. 31.7 ± 26.2 days, $p < 0.001$), more with congestive heart failure (42.0% vs. 25.4%, $p = 0.020$), and had different distribution of the types of acute respiratory failure ($p = 0.037$) (Table 2).

## Comparisons of physiological parameters and WS

Table 3 compares the physiological parameters related to the assessments during the weaning process and WSs between the success and failure groups. Univariate analysis

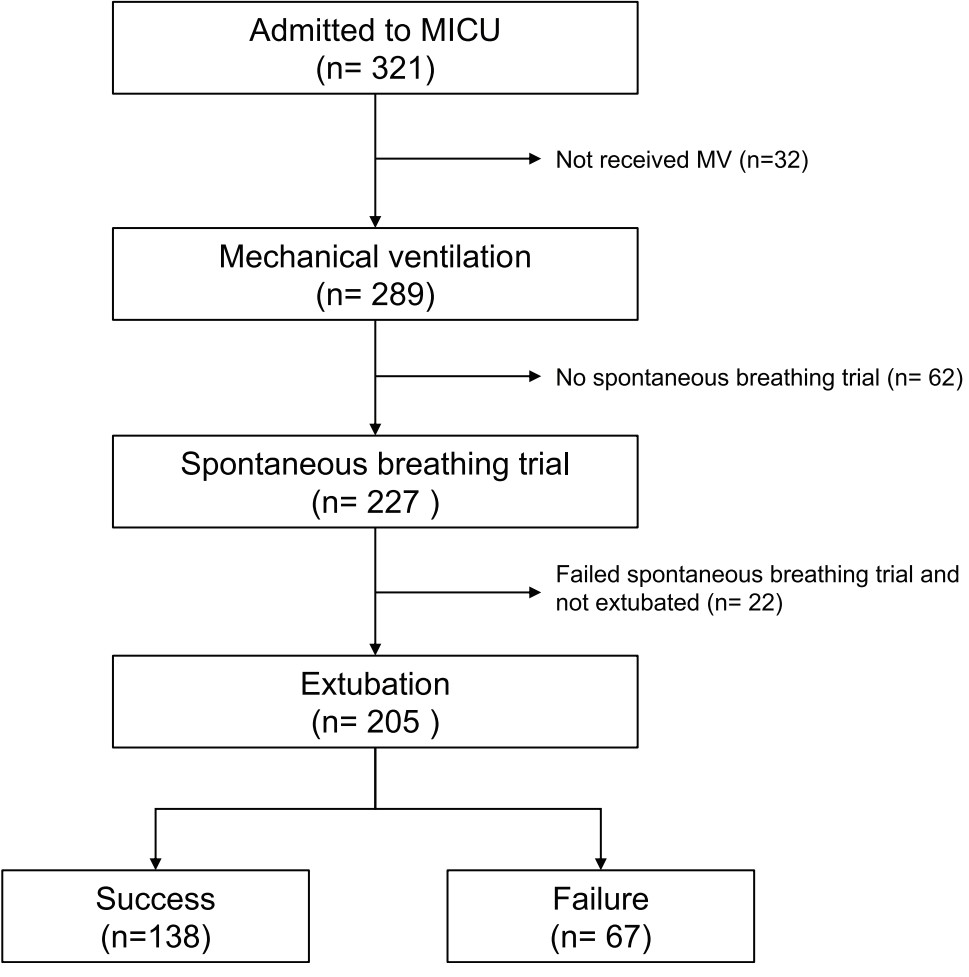

**Figure 1 Flow diagram of the patients included in this study.** Detailed statement of the numbers of inclusion and exclusion for this study.

showed significant differences in maximum inspiratory pressure ($p = 0.001$), maximum expiratory pressure ($p = 0.005$), spontaneous tidal volume ($p < 0.001$), respiratory rate ($p < 0.001$), rapid shallow breathing index ($p < 0.001$), average WS ($p < 0.001$) and three of the components of the scoring system, including weaning parameters ($p < 0.001$), neuromuscular ($p = 0.039$) and wakefulness ($p = 0.001$) assessments between the two groups. More patients in the weaning success had both RSBI < 105 and WS = 0 than those in the failure group (59.4% vs. 22.4%, $p < 0.001$).

The results of multivariate logistic regression analysis for the factors associated with extubation success in the included patients are summarized in Table 4. A WEANSNOW Score ≥ 1 (OR = 2.880 (95% CI [1.291–6.426]), $p = 0.010$) and intubation duration >21 days (OR = 7.752 (95% CI [3.560–16.879]), $p < 0.001$) were independently associated with an increased probability of extubation failure.

For sensitivity analysis, we also separated the item "wakefulness" from the WS score in the multivariate analysis. Logistic regression showed that the "total WS minus the wakefulness score ≥1" remained a significant associating factor for extubation failure

**Table 2 Comparisons of demographic and clinical characteristics of the patients with successful and failed weaning attempts.**

| Variable | Total patients (n = 205) | Weaning success (n = 138) | Weaning failure (n = 67) | p-Value |
|---|---|---|---|---|
| Age, years | 68.5 ± 14.0 (24–97) | 68.1 ± 13.7 (34–93) | 69.1 ± 14.8 (24–97) | 0.637 |
| Male, n (%) | 124 (60.5%) | 85 (61.6%) | 39 (58.2%) | 0.642 |
| Duration of intubation, days | 17.9 ± 20.1 (2–108) | 11.2 ± 11.6 (2–67) | 31.7 ± 26.2 (2–108) | <0.001 |
| Main comorbidity, n (%) | | | | |
| Congestive heart failure | 75 (36.6%) | 58 (42.0%) | 17 (25.4%) | 0.020 |
| Chronic obstructive pulmonary disease | 43 (21.0%) | 28 (20.3%) | 15 (22.4%) | 0.729 |
| Chronic kidney disease | 69 (33.7%) | 45 (32.6%) | 24 (35.8%) | 0.648 |
| Hypertension | 95 (46.3%) | 65 (47.1%) | 30 (44.8%) | 0.754 |
| Diabetes | 52 (25.4%) | 35 (25.4%) | 17 (25.4%) | 0.999 |
| Malignancy | 92 (44.9%) | 57 (41.3%) | 35 (52.2%) | 0.140 |
| Type of respiratory failure, n (%) | | | | |
| Type I | 57 (27.8%) | 41 (29.7%) | 16 (23.9%) | 0.037 |
| Type II | 86 (42.0%) | 64 (46.4%) | 22 (32.8%) | |
| Type III | 9 (4.4%) | 4 (2.9%) | 5 (7.5%) | |
| Type IV | 53 (25.9%) | 29 (21.0%) | 24 (35.8%) | |
| APACHE II score on admission | 21.3 ± 8.1 | 21.4 ± 8.1 | 21.1 ± 8.1 | 0.860 |

Note:
APACHE, acute physiology and chronic health evaluation.

(OR = 3.552 (95% CI [1.548–8.150]), $p$ = 0.003), along with intubation duration >21 days (OR = 7.862 (95% CI [3.507–17.623]), $p$ < 0.001). In contrast, wakefulness was not independently associated with extubation failure in this study group (OR = 1.372 (95% CI [0.630–2.991]), $p$ = 0.426) (See Table 1S of File S2).

## DISCUSSION

In this study, we developed a composite assessment score for the preparedness of weaning to study its association with the outcomes of weaning in patients with acute respiratory failure receiving MV. The results showed that the composite WS, intended to be used in a "bundle" score, was associated with weaning success when the score was 0 independently of the commonly used rapid shallow breathing index.

Successful extubation is traditionally predicted based on physiological assessments and calculated parameters such as the rapid shallow breathing index (*Yang & Tobin, 1991*). Other prediction models include APACHE II score (*Afessa, Hogans & Murphy, 1999*), CROP index (*Yang & Tobin, 1991*), relative inspiratory effort (*Kline et al., 1987*), alveolar-arterial oxygen tension gradient + blood urea nitrogen + gender score (A + B + G) (*Scheinhorn et al., 1995*), new weaning index based on ventilatory endurance and the efficiency of gas exchange (*Jabour et al., 1991*), and Integrated Weaning Index (*Nemer et al., 2009*). Several studies have reported that commonly used weaning indices have suboptimal prediction power for successful liberation from MV (*MacIntyre et al., 2001*; *Kacmarek, Stoller & Heuer, 2016*). These indices are commonly implemented in a clinical setting, are performed by respiratory therapists and focus more on ventilation function

**Table 3 Comparison of physiological parameters and WEANSNOW score.**

| Variable | Total patients (n = 205) | Weaning success (n = 138) | Weaning failure (n = 67) | p-Value |
|---|---|---|---|---|
| MIP, cm $H_2O$ | −37.1 ± 11.4 | −38.9 ± 11.1 | −33.4 ± 11.2 | 0.001 |
| MEP, cm $H_2O$ | 43.4 ± 16.8 | 45.7 ± 16.9 | 38.7 ± 15.6 | 0.005 |
| $V_T$, mL | 395.4 ± 150.0 | 429.2 ± 150.8 | 325.9 ± 122.8 | <0.001 |
| $V_E$, L/min | 8.3 ± 3.2 | 8.4 ± 2.8 | 8.1 ± 3.7 | 0.567 |
| RR | 21.7 ± 6.5 | 20.3 ± 6.0 | 24.7 ± 6.5 | <0.001 |
| RSBI | 68.3 ± 49.4 | 56.3 ± 35.1 | 93.9 ± 63.8 | <0.001 |
| WEANSNOW component | | | | |
| Weaning profile, passed | 148 (72.2%) | 114 (82.6%) | 33 (49.3%) | <0.001 |
| Endotracheal tube, passed | 201 (98.0%) | 137 (99.3%) | 64 (95.5%) | 0.103 |
| Arterial blood gas analysis, passed | 205 (100%) | 138 (100%) | 67 (100%) | 1.000 |
| Nutrition, passed | 201 (98.0%) | 135 (97.8%) | 66 (98.5%) | 1.000 |
| Secretions, passed | 194 (94.6%) | 130 (94.2%) | 64 (95.5%) | 0.758 |
| Neuromuscular, passed | 198 (96.6%) | 136 (98.6%) | 62 (92.5%) | 0.039 |
| Obstruction of the airway, passed | 198 (96.6%) | 135 (97.8%) | 63 (94.0%) | 0.219 |
| Wakefulness, passed | 137 (66.8%) | 103 (74.6%) | 33 (49.3%) | 0.001 |
| WEANSNOW score, average | 0.75 ± 0.83 | 0.51 ± 0.71 | 1.22 ± 0.85 | <0.001 |
| WEANSNOW score | | | | |
| 0 | 98 (47.8%) | 83 (60.1%) | 15 (22.4%) | <0.001 |
| ≥1 | 107 (52.2%) | 57 (40.7%) | 50 (76.9%) | |
| RSBI < 105 and WEANSNOW score = 0 | 97 (47.3%) | 82 (59.4) | 15 (22.4%) | <0.001 |

**Note:**
MIP, maximum inspiratory pressure; MEP, maximum expiratory pressure; $V_T$, tidal volume; $V_E$, minute ventilation; RR, respiratory rate; RSBI, rapid shallow breathing index.

**Table 4 Multivariate logistic regression analysis of the factors and variables associated with weaning success.**

| Variable | N (%) | OR | 95% CI | p-Value |
|---|---|---|---|---|
| Heart failure | 75 (36.6%) | 0.609 | [0.276–1.344] | 0.219 |
| Type 1 or Type 2 respiratory failure | 143 (69.8%) | 0.533 | [0.252–1.128] | 0.100 |
| Intubation days > 21 | 53 (25.9%) | 7.752 | [3.560–16.879] | <0.001 |
| MIP ≤ −20 cm $H_2O$ | 189 (92.2%) | 0.559 | [0.143–2.189] | 0.403 |
| MEP ≥ 30 cm $H_2O$ | 157 (76.6%) | 0.996 | [0.405–2.449] | 0.993 |
| RSBI < 105 | 26 (13.3%) | 0.390 | [0.135–1.126] | 0.082 |
| WEANSNOW score ≥ 1 | 107 (52.2%) | 2.880 | [1.291–6.426] | 0.010 |

**Note:**
MIP, maximum inspiratory pressure; MEP, maximum expiratory pressure; RSBI, rapid shallow breathing index; OR, odds ratio; CI, confidence interval.

and less on airway condition. Previous reports have suggested that a substantial proportion of patients who fail extubation after a successful spontaneous breathing trial should be defined as "airway failure", especially those who are reintubated within 48 h after extubation. We also used 48 h to define extubation success in our analysis. We found that most of the patients who were prepared for extubation had a WS of 0, suggesting that the
clinicians were also confident about the preparedness for extubation. Therefore, we did not perform the receiver operating characteristic curve analysis to compare the accuracy with other weaning indices.

We also focused on components other than direct measurements of respiratory mechanics, thereby emphasizing the importance of adequately maintained natural airway function. The effect of an adequately sized artificial airway on the work of breathing is controversial (*Straus et al., 1998*). However, endotracheal tubes, which are partially obstructed by airway secretions, may not show abnormal findings by measuring weaning parameters, and may, therefore, also significantly increase the work of breathing after extubation, potentially resulting in airway failure requiring re-intubation. The size of the endotracheal tube can also affect the results of weaning trials. In critically ill patients, a decrease in internal diameter and an increase in minute ventilation increase the work of breathing. Adults have a critical increase in workload when the tube has an internal diameter of less than 7 mm (*Sharar, 1995*). Biofilm buildup on the inside of the endotracheal tube can cause dramatic increases in airway resistance, especially among infants and children (*Mietto et al., 2014*). Furthermore, the presence of upper airway obstructions or laryngeal edema may be detected by a cuff leak test showing diminished gas leak around the endotracheal tube with positive pressure breaths (*Wratney & Cheifetz, 2007*). Arterial gas analysis, nutritional impairment (*McClave et al., 2009*) and neuromuscular impairment are considered to affect ventilation function related to endurance, which may not be recognized by measuring weaning parameters alone. Finally, we found that wakefulness was significantly different between the success and failure groups, suggesting that this component contributes to both airway failure and ventilation failure, ultimately resulting in re-intubation.

The WEANSNOW bundle may assist in optimizing the preparedness of weaning-extubation and the WS may be a useful assessment tool to understand the condition of patients with acute respiratory failure more thoroughly. Although for each component, the majority of patients passed the criterion, our data also showed that only 48% had a total score of zero. Therefore, we would suggest that a bundled assessment might still be needed to remind the clinician about the risk of extubation failure. All of the clinical conditions related to the individual components of the WS can be improved before weaning-extubation if the patient is at high risk of failure. Multivariate regression analysis (Table 4) showed that the other variables used to predict the success of weaning-extubation, such as comorbidities, the reason for intubation, age and sex, are non-modifiable and therefore only serve as predictors. Even though the components of the WS can be improved before attempting extubation, patients who are already intubated with an endotracheal tube with a smaller internal diameter or those with chronic pulmonary and airway diseases such as chronic obstructive pulmonary disease that reduce the WS may not benefit from the WS assessment. Otherwise, the implementation of this bundle assessment to both predict and guide patient care might be considered.

There are several limitations to this study. The author decided each criterion for the individual components based on the clinician experience-based consensus as a lack of evidence-based generation of the WEANSNOW criteria. Although the result of

multivariate regression showed the significance of WEANSNOW as a bundled assessment, the individual component might need further evaluation in future studies employing prospective inclusion of a sufficient number of patients of different characteristics and physiologic functions, which is beyond the scope of this retrospective study. Also, as our searching showed, most of the articles describing WEANSNOW appear to based on expert experiences and opinions provided in book chapters as a guide to the process in professional practices regarding ventilator weaning and extubation. Nevertheless, the lack of clinical evidence was also one of our original motivations to analyze the usability of this checklist-based assessment. Further evidence-based evaluation of the practice is necessary. Some patient characteristics were difference between the success and failure groups, such as duration of intubation, congestive heart failure as a comorbidity and the type of respiratory failure as a reason to intubate. Despite that these characteristics were included in the multi-variate analysis, further prospective randomized studies are needed to address these potential factors, such as stratification of patients with different co-morbidity, duration of intubation and type of respiratory failure. This study only included intubated patients in the medical ICUs of a single medical center. The patients underwent weaning and extubation mainly based on the decisions of the physician and the nurses followed standardized nursing care procedures. Our findings may not be generalizable to ICUs that have implemented weaning-extubation processes by respiratory therapists, or those that use similar checklist-based assessments and management strategies before proceeding with weaning-extubation. Besides, typical patients treated at the surgical ICU are mechanically ventilated for a brief period after surgery; therefore, the WS needs to be validated in the non-medical population of ICU patients. The design of this study designated those with post-extubation NIV use as failure group (*Rothaar & Epstein, 2003*). Recently, the potential of earlier extubation with immediate respiratory support using noninvasive ventilation has been suggested (*Yeung et al., 2018*). As the ICUs in this study did not routinely apply NIV for extubated patients unless there was evidence of worsening respiratory conditions requiring MV support, our study design excluded those patients who required either invasive or non-invasive ventilation within 2 days after extubation. Our study findings might not be generalized to the ICUs that routinely apply NIV proactively after extubation. However, it would be an interesting question whether or not patients with higher WEANSNOW scores can be successfully extubated with NIV as a step down therapy, and prospective studies are needed. Furthermore, regarding the performance of this ICU, only 51.2% were transferred out of the ICU after successful liberation from MV. These were medical ICUs that most of the patients have impaired gas exchange rather than merely post-procedural liberation of MV. The practice of physicians may have especially affected our determination of extubation outcomes as we defined successful extubation as unassisted spontaneous breathing maintained for at least 48 h. Therefore, the contribution of ICU performance to WEANSNOW would also need further investigation. This was a retrospective analysis, and the WS was based on information retrieved from the electronic medical records of the hospital. The information needed to decide whether or not the operational criteria of the

WEANSNOW bundle checklist had been fulfilled did not necessarily assure actual optimal preparation in that regard. We included all of the patients who experienced an actual extubation attempt; therefore, the study allowed more than one spontaneous breathing trial before extubation, despite that only the assessment data closest to the time of extubation would be used. Our inclusion of only those patients who experience extubation would preclude those who did not undergo extubation regardless of the scoring of WEANSNOW. An auditing process may be more suitable for the WS. This study did not include a prospective investigation of predictability using a validation cohort of patients with the same inclusion-exclusion criteria. Physicians based on their clinical judgment made the decisions for extubation and re-intubation; however, their decision on whether or not to extubate may have also been based on clinical judgment related to the WEANSNOW components, and therefore the patients with a high probability of extubation failure may have remained intubated until their clinical condition improved. The post-extubation management may not have been standardized so that the decision to re-intubate may also have been affected by the clinicians. For the understanding of the impact of this pre-extubation practice with bundle checklist such as WEANSNOW, a standardized post-extubation care process till ICU discharge or re-institution of MV might be needed.

## CONCLUSIONS

In conclusion, assessing the pre-extubation status of intubated patients using the WS might provide valuable insights into extubation failure in patients in a medical ICU for acute respiratory failure. As the individual components can be improved by active management, this bundle-based approach may also provide goals to optimize the patients' condition before extubation to achieve better outcomes regarding liberation from MV. Further prospective studies are warranted to elucidate the practice of assessing weaning preparedness.

### Funding
The authors received no funding for this work.

### Competing Interests
The authors declare that they have no competing interests.

### Author Contributions
- Feng-Ching Lin conceived and designed the experiments, performed the experiments, analyzed the data, prepared figures and/or tables, authored or reviewed drafts of the paper, and approved the final draft.
- Yao-Wen Kuo performed the experiments, analyzed the data, authored or reviewed drafts of the paper, and approved the final draft.

- Jih-Shuin Jerng conceived and designed the experiments, performed the experiments, analyzed the data, prepared figures and/or tables, authored or reviewed drafts of the paper, and approved the final draft.
- Huey-Dong Wu conceived and designed the experiments, authored or reviewed drafts of the paper, and approved the final draft.

## Human Ethics

The following information was supplied relating to ethical approvals (i.e., approving body and any reference numbers):

The Institutional Research Ethics Committee of National Taiwan University Hospital approved this study (201808101RINA) and waived the need for informed consent from the patients.

## Data Availability

The raw measurements are available in the Supplemental File.

## Supplemental Information

Supplemental information for this article can be found online at http://dx.doi.org/10.7717/peerj.8973#supplemental-information.

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
