# Peer review of "Association of weaning preparedness with extubation outcome of mechanically ventilated patients in medical intensive care units: a retrospective analysis"

_PeerJ, doi:10.7717/peerj.8973_

## Round 0.1 · original submission · Major Revisions

· Academic Editor

Major Revisions

Please see the comments from both reviewers, in particular those of reviewer number 1. And please pay careful attention to the limitations related to experimental design and conclusions. Any elaboration that is not possible must be addressed as a limitation.

As it pertains to both reviewers and references please make sure to add additional references to support your manuscript/research that are both more current and represent peer-reviewed published manuscripts.

·

Basic reporting

For the most part, the article meets the standards.
The literature references use textbooks on some critical parts. I would recommend journal articles for these, even if only in addition, as textbooks are not peer reviewed and therefore not up-to-date per se.

Experimental design

The article falls within the aims and scope.
All technical and ethical standards were followed.
However, there is quite a difference in the patient characteristics in both groups that could lead to quite a big bias, and leading to wrong conclusions.
A more than tripling of the ventilation duration in the 'unsuccessful' group can lead to unacceptable bias. A better case-control or propensity-matching strategy should be (at least) considered.
Also the WEANS NOW criteria are not generally standardised. It would be better if the authors would elaborate on why the criteria were chosen 'as is', instead of just 'after a brainstorm session'. Especially as most of the criteria shown were not significantly attributing to the predictive value. Also, the verbal component of the GCS is a bit controversial for intubated patients.A bit more explanation on their standard would be advisable.
It is not stated whether or not more than one spontaneous breathing trials were (allowed to be) performed before actual weaning. It probably is, as the numbers in line 154 and 164 do not concur otherwise. This is an extra possible bit of bias.
Lastly, some more characteristics of the performance of the ICU would be appreciated. What is their current weaning failure? Then the conclusion would show whether or not the WEANSNOW could contribute to the performance.

Validity of the findings

See also 2.
All raw material is provided. The conclusions are quite a bit stronger than can be supported by the given material. The difference in patient characteristics is high between both groups. There are many reasons for higher failure to wean in longer ventiation duration, like ventilator associated pneumonia and muscle wasting to name only two. These can't be ignored by stating that the characteristics at the time of admission were comparable. Also the percentage of congestive heart failure was significantly higher in the unsuccessul group.
Moreover, even so the prediction cut off is 1, while the average of the successful group is about 0.54, and the unsuccessful is 1.18, a very small difference for a 8-point scale, and no patient is above 3 (and one of the 3 patients that scored three had a succesfull extubation).

Additional comments

I would think that with some elaboration on certain parts, it could be published. But the conclusions would probably quite be a bit less strong than it is now.

Reviewer 2 ·

Basic reporting

1.1. Lines 48-49 reference an article over a decade old which describes different weaning processes. This seems like an arbitrary citation as this terminology is not commonly used internationally. Since differentiating between the three processes does not seem to be significant to the article, I suggest removing the sentence entirely.
1.2. Several studies referenced are decades old and were not landmark studies in the field. It is important to build upon the current evidence base; updating references is highly recommended.
1.3. Line 190: Respiratory therapists are mentioned here as the people who perform weaning assessments but it is unclear how that information is relevant. Is the WEANSNOW evaluation not performed by respiratory therapists?
1.3.1. Are respiratory therapists commonly used in this geographical area? Having that bedside expertise (or lack of) may contribute to weaning readiness and success/failure. Clarification recommended.
1.4. Table 3 and Table 4: RSBI > 105 is again referenced (see 3.1). RSBI should be less than 105 to align with the standard practice, unless the authors here are proposing something new.

Experimental design

2.1. It is unclear why participants were enrolled at a 2:1 ratio for success to failure. Was it only to get a larger sample size or some other reason?

Validity of the findings

3.1. Lines 169-172: An increased WEANSNOW score and RSBI > 105 should correlate with a decreased probability of extubation success. Is this a typo? RSBI < 105 is known to be the standard for assessing extubation readiness. How the article is currently written (suggesting RSBI > 105) would contradict a significant amount of published literature.
3.2. Wakefulness seems to have substantial influence on patient performance. Should this be assessed independently or does research already exist? How valid is the WEANSNOW score if wakefulness is removed?

Additional comments

4.1. Lines 66-67 use the word evaluation twice in one sentence which may be confusing. Consider revision.
4.2. Line 165: RSBI is listed as PSBI – needs typo correction
4.3. Line 225: Use smaller internal diameter as it is more universal language.
4.4. The focus on developing a weaning assessment bundle which accounts for airway condition, not just mechanics, is a worthy venture and is overdue.
4.5. Several limitations are identified and thoroughly addressed.

---

## Round 0.2 · Minor Revisions

· Academic Editor

Minor Revisions

Please address the minor revisions request as noted by the current review. In particular please consider the reviewers suggestions about how strongly you report your conclusion in the discussion section.

·

Basic reporting

Some typos. nothing major:
271-272: Typo, it should be “…WEANS NOW appear to be based…”.
295: Typo: experience should be experienced.

Perhaps some elucidation:
80: The Institutional Research Ethics Committee A of the National Taiwan University Hospital: is this correct? Is the Research Ethics Committee “A”? Or is it a typo? No need to change it per se if it is correct.

274-275: “…was also one of our original intentions…”. Intentions is probably not a correct word here, shouldn’t it be ‘reasons’ or ‘motivations’?

Experimental design

171-173: what is the reason for mentioning the use of NIV? Is NIV to be viewed as an extubation-failure, or as a normal ‘step-down’ therapy or adjunct after MV? In my opinion it doesn’t really clarify anything, so I would just leave it out, and use a dichotomous extubation success or failure (unless WEANS NOW would predict usefulness of NIV after extubation, but I reckon this can’t be concluded from the current data). You talk about NIV as a step down in lines 283 and further, but the relation between the predictability of success of extubation using WEANS NOW with or without NIV is not researched here.

283-285: As stated before, the WEANS NOW wasn’t used with prediction of succes with or without NIV. I would rephrase, that it would be an interesting question whether or not patients with higher WEANS NOW scores can be succesfully extubated with NIV as a step down therapy.

287-288: I would elaborate on when a death or a transfer in a ventilator dependent state is viewed as still being a succesful extubation.You do this in lines 308-, but I think this should be done here, or it raises questions about almost 50% dying or needing ventilation after a successful extubation.

Validity of the findings

The conclusions reached are still a bit stronger than I would have reached myself. The use of WEANS NOW is not as strong a predictor as is suggested. The current conclusion is correct, but the discussion suggests something else:
263-264: “Otherwise, we strongly recommend…”

This seems like a bit of a strong recommendation, especially in the Discussion part of the article.

Also, I would address the differences in patient characteristics a little bit more in the Discussion. It is mentioned in the results, but it is not discussed as being a limitation (or not), or the possible effects on the interpretation.

Additional comments

The revisions so far have improved the article significantly. There is some difference in point of view with regards to the eventual conclusion. But with certain caveats it is defendable.
Most importantly I would like to see some discussion over the (possible extent or lack thereof of the) difference in patient characteristics.

---

## Round 0.3 · accepted · Accept

· Academic Editor

Accept

Thank you for your persistence and professional perseverance in making the revisions as outlined by the last review recomendations. I am in agreement with the reviewers that there has been substantial improvement in the manuscript.